# Perspectives of stakeholders on barriers to COVID-19 protective behaviors adherence and vaccination among Myanmar migrant workers in southern Thailand: A qualitative study

**Hein Htet[1,2], Wit Wichaidit[1]\*, Aungkana Chuaychai[3], Tiida Sottiyotin[3], Kyaw Ko Ko Htet[4], Hutcha Sriplung[1], Virasakdi Chongsuvivatwong[1]**

**1** Department of Epidemiology, Faculty of Medicine, Prince of Songkla University, Hat Yai, Songkhla Province, Thailand, **2** Department of Preventive and Social Medicine, University of Medicine (Taunggyi), Ministry of Health, Myanmar, **3** Department of Pharmaceutical Care, School of Pharmacy, Walailak University, Nakhon Si Thammarat Province, Thailand, **4** Independent investigator

\* wit.w@psu.ac.th

## Abstract

Studies have been conducted on migrant health during the COVID-19 pandemic. However, in-depth information is scarce regarding the barriers to preventing COVID-19 in this vulnerable population. The objective of the study is to explore the barriers to COVID-19 protective behaviors adherence and vaccination among Myanmar migrant workers in Thailand. We conducted an interview-based qualitative study among 7 migrants from Myanmar, 6 Thai employers, and 9 Thai healthcare providers in the cities of Hat Yai and Pattani in Southern Thailand. We recruited participants by purposive sampling. We conducted in-depth interviews in-person or via telephone in Thai or Burmese language, transcribed the interview, and conducted thematic analysis. Regarding adherence to COVID-19 protective behaviors, two themes emerged: lifestyle and habit-related barriers, and non-vaccine supply chain management barriers. Regarding COVID-19 vaccination, three common themes emerged: fear, barriers related to health education and health promotion, and vaccine supply chain management. Supply chain management was a common theme in both domains. However, each domain also had additional themes. Our study contributed empirical findings that could be of interest to stakeholders in migrant health. However, limitations regarding the generalizability of the findings and social desirability should be considered in the interpretation of the findings.

## Introduction

Globally, there are 281 million international migrants, including 169 million migrant workers [1]. There are different categories of migrants, including economic migrants, refugees, asylum seekers, and internally displaced persons [1]. Health-related challenges among migrants in host countries include cultural, structural, financial and language barriers [2]. During the

**Data availability statement:** All relevant data are within the manuscript and its Supporting Information files.

**Funding:** The author(s) received no specific funding for this work.

**Competing interests:** The authors have declared that no competing interests exist

COVID-19 pandemic, migrants and ethnic minorities were disproportionately affected by stigma and discrimination [3], as well as being blamed for spreading infections [4,5]. In many countries, migrants were excluded from the COVID-19 relief policy measures, especially if they were undocumented and asylum seekers [6]. Thailand, an upper-middle income country in South East Asia, is one of the top ten destination countries for international migrants in the WHO South East Asia Region [7]. The majority of migrants in Thailand come from neighboring Myanmar with 1.65 million registered workers in 2021 according to the statistics from the Department of Employment, Ministry of Labour, Thailand [8], most of whom work in low-skilled jobs such as construction, factories, and fisheries [9].

Thailand experienced five distinct waves of COVID-19 between 2020 and 2022, where the fifth wave of COVID-19 occurred in late 2022, and included the emergence of more transmissible variants like Alpha and Delta [10]. As of 13 April 2024, Thailand had a cumulative total of 4,770,149 confirmed cases with 34,586 deaths [11]. During the pandemic situation in Thailand, Myanmar migrants experienced higher infection rates especially in the second wave [12–16], faced a shortage of personal protective equipment (e.g., face masks, hand sanitizers, and soaps [17,18], faced challenges in adhering to COVID-19 protective measures [19], and faced barriers in healthcare access due to structural, financial, language, and communication difficulties [20]. The pandemic also exacerbated the discrimination against migrants [21].

Public health interventions to prevent and control COVID-19 have included hand hygiene, face covering use, social distancing in public places, isolation, travel restrictions, vaccination, etc. [22–25]. Different interventions affect disease transmission in different ways [26–28], and a comprehensive strategy is deemed needed in order to effectively control COVID-19 infection [29].

Thailand's Ministry of Public Health Thailand formulated the National COVID-19 Strategic Plan, with relevant public health measures and social measures [30]. On 28 February 2021 [31], a nationwide government-funded free COVID-19 vaccination program was also launched [16]. The program prioritized frontline health care personnel, people with chronic diseases, and elderly people during the initial period [16], before expanding in the last quarter of 2021 to include non-Thai migrant workers and their families regardless of legal status [16]. However, Myanmar migrants in Thailand were found to have low COVID-19 vaccination rates due to several barriers, including the exclusion from the vaccination program, language barriers, and financial constraints. These challenges are further compounded by previous studies conducted in Southern Thailand, which highlighted unsatisfactory health-seeking behaviors among Myanmar seafarers in Pattani province [32], as well as factory, construction, and rubber trapping workers in Songkhla province [33].

Although studies have been conducted regarding COVID-19 vaccination among migrants, studies on migrant health in Thailand only included the perspectives of the migrants but not their Thai employers or healthcare workers [20] who coordinate and provide vaccination and other resources for the migrants. Recent research on COVID-19's impact on vulnerable migrant communities in Thailand incorporated diverse viewpoints from various stakeholders, including representatives of migrant advocacy organizations, NGOs, and community-based support groups [34]. Such data can provide valuable insights for relevant stakeholders that enable planning and preparations for future infection control practices and crises. Building upon this foundation, our qualitative study aims to provide a comprehensive understanding of the barriers to COVID-19 protective behaviors adherence and vaccination among Myanmar migrant workers in Thailand according to the perspectives of three important stakeholders for migrant health: migrant workers, employers, and healthcare providers. This multi-stakeholder approach will generate valuable insights to inform future crisis preparedness and infection control strategies.

## Methods

### Study design and setting

This study was a descriptive qualitative study conducted from the 1st of September 2022 to the 24th of January 2023 during the COVID-19 pandemic. During the COVID-19 pandemic (2020–2022), the industries in Southern Thailand were transiently affected, and the government issued several non-pharmacological measures, including social distancing, compulsory mask wearing, travel restrictions, and workplace disinfection, to prevent further spread among the workers. In late 2021, the government offered vaccinations to migrant workers free of charge as part of an inclusive vaccine policy [35].

Study areas included the cities of Hat Yai city of Songkhla province and Pattani city of Pattani province in southern Thailand. Songkhla and Pattani provinces are major industrial commercial area of the Southern Thailand with many factories that process rubber, wood and seafood as well as numerous construction and fishery sites [36]. Many Myanmar migrant laborers live in these provinces, with different ethnic groups, religions, and cultures. According to the 2021 Thailand Employment Statistics, there were 19,810 and 4,122 legal Myanmar migrant workers granted by cabinet resolution in Songkhla and Pattani provinces, respectively [37].

### Study participants

Twenty-two stakeholders were interviewed, including Myanmar migrants (n = 7 persons), Thai employers (n = 6 persons), and Thai healthcare providers (n = 9 persons). Migrants were enrolled if the following predetermined inclusion criteria was met: (i) Myanmar Nationality; (ii) age at least 18 years old; (iii) residing or working in the study area for at least six months, and; (iv) able to communicate in Burmese. Migrants with the following criteria were excluded from the study: (i) those with mental or auditory problems; (ii) those who received COVID vaccination outside of Thailand. Inclusion criteria for employers and health care provider were (i) Thai nationality (ii) Government or Non-government officers whose work was related to COVID-19 vaccination services for migrants within the study area for at least six months; (iii) Owners, managers, supervisors in-charges, safety officers, etc., who were employing Myanmar migrants within the study area for at least six months. Their exclusion criteria were (i) employers with no experience of employing Myanmar migrants; (ii) healthcare providers who were working at facilities with migrant exclusionary policy. Participants were selected through purposive sampling by the research team and local key informants to ensure a diverse range of perspectives within each stakeholder group. Myanmar migrants were identified with the help of the key informants from the Migrant Workers Right Network (MWRN) and the Stella Maris Organization. These key informants were Myanmar Nationalities who had been working in the study area for more than 10 years. They could well acquaint with the Myanmar migrant population within Hat Yai and Pattani and could provide information on places of worksite and residence, types of occupations, legal status, approximate population sizes, and their lifestyles. For Thai employers and Thai healthcare providers, they were also identified by the Thai research coordinator and invited via official request letters, emails, or phone calls and invited to participate in individual in-depth interviews.

### Study instrument

The semi-structured in-depth interviews guides were developed based on the previous COVID-qualitative studies. Three separate question guidelines were used for the migrant workers, the employers, and the health care providers. The interview guides included open-ended questions to explore the main issues related to potential challenges for adherence to

COVID-19 protective measures (e.g., In your opinion, what do you consider to be your main limitations or barriers to adhere or follow COVID-19 protective measures in your workplace? Would you mind telling us about these barriers in detail?) and vaccination (Example: "In your opinion, what do you consider to be the main barriers, if there are any, to COVID-19 vaccination for you and why?") from the aspects of Myanmar migrant workers, Thai health care providers, and Thai employers. The interview guides were first drafted in English and then translated into Burmese (for Myanmar migrant workers) and into Thai (for Thai health care providers, Thai employers) and validated before implementation of the main interviews upon discussion with experts. Prior to the main interviews, the interview guides were piloted and iteratively revised based on feedback from the participants.

## Data collection

After receiving ethical approval and permission from the participants, the research team scheduled in-depth interviews upon their feasible date and time. Appointments for interviews were scheduled 1 or 2 weeks prior to the actual interview date. Due to the COVID-19 pandemic, individual interviews were conducted either in person or via telephone, by three experienced investigators with prior qualitative research experience. We conducted all in-person interviews at the participant's workplace. The primary investigator (HH), a Myanmar national, conducted interviews with migrants in Burmese, and the two Thai co-investigators (AC and TS) interviewed with both employers and healthcare providers in Thai. We matched the nationality of the interviewers and participants in order to reduce potential language barriers, manage cultural nuances, and enable more effective probing. At the time of the study, HH was a male PhD student with a background as a public health physician. During the interview, HH made attempts to be conversational and build rapport with the participants in order to lower the potential social desirability and response acquiescence. The Thai co-investigators were female university lecturers with graduate-level education. The participants and the members of the research team were never acquainted with one another prior to data collection. None of the research team members had experience working or advocating on Myanmar migrant issues prior to the study. Each interview lasted between 45 and 60 minutes, with an explanation of the purpose and procedure of the study to the participants before the start of the interviews. No individuals other than the interviewer and the participant were present at all of the interviews. Following each interview, a brief interview summary was provided to all participants to verify the accuracy of the content and key messages. The participants did not provide us with any feedback after the interview summary. We did not make field notes during and/or after the interviews. We did not conduct any repeat interviews.

## Data management and analysis

All interviews were audio recorded with consent and transcribed verbatim. All transcripts were carefully reviewed by study investigators to check the accuracy with the identification of new concepts and the assessment of data saturation levels. For in-depth interviews of the migrant workers in Burmese, the primary investigator (HH) who spoke Burmese as a native language transcribed the interview to Burmese texts, then used an artificial intelligence system (ChatGPT) to translate the Burmese texts into English. HH then manually checked the translations and corrected the parts deemed to be inaccurate. For in-depth interviews of the employers and healthcare providers in Thai, the investigators hired native Thai speakers via a popular freelance website to transcribe the interview into Thai texts. Investigators then used a machine-assisted translation tool (Google Translate) to translate the Thai texts into English texts. A co-investigator and the corresponding author (WW) who spoke Thai as a native

language, manually checked the translations and corrected the parts deemed to be inaccurate. The decision to use ChatGPT to translate Burmese texts was made after HH made an assessment of both ChatGPT and Google Translate and found that ChatGPT had a higher accuracy and a better ability to manage cultural nuances. The decision to use Google Translate for the Thai texts was based on WW's prior experiences with the validation of translated texts.

Investigators analyzed the qualitative data in the English translations. The transcribed data were inductively explored manually, using thematic analysis in accordance with the research objective. The analytical steps were as follows: first, open coding, defining as many codes as needed to describe all aspects of the content; second the codes were categorized to create themes and sub themes, all leading to an explanation. Two investigators (HH and WW) identified relevant text segments in the translated transcripts based on consensus. The two investigators then separately identified codes, sub-themes, and themes, based on the frequency of occurrence and significance, relevance to research objectives, cultural and contextual sensitivity [38–40]. Then, the two investigators met after the completion of the process to cross-check the findings, discuss discrepancies, and finalize the themes, sub-themes, and codes based on consensus. The two investigators then asked a third investigator, a subject specialist in infectious diseases (VC) to verify the interpretation and conclusions. VC informed the investigators that he concurred with the interpretation and the conclusion and that he had no further comments. The investigators then made coding trees to summarize the findings.

### Ethical considerations

This qualitative study was approved by the Human Research Ethics Committee of the Faculty of Medicine, Prince of Songkla University, Hat Yai, Thailand (approval number: REC. 65 – 071 – 18 – 1). This study was also conducted in full compliance with the COVID-19 preventive guidelines. Prior to participation, only verbal consent was obtained from migrants due to the sensitive nature and confidentiality of their immigration status, whereas written informed consent or verbal consent was obtained from Thai health care providers and Thai employers.

## Results

### Study participants

All potential participants agreed to participate in the study. A total of 7 Myanmar migrant workers (three worked in factories, two seafarers, and two construction workers), 6 Thai employers, and 9 Thai healthcare workers participated in our in-depth interviews. All participants resided in Songkhla or Pattani Provinces at the time of the interview. Their relevant background information is summarized in Table 1.

### Key themes related to the barriers to adherence to COVID-19 preventive behaviors

With regards to adherence to COVID-19 prevention behaviors, each of the three stakeholder groups revealed several key themes (Table 2). The most common theme reported by all three groups was lifestyle and habit-related barriers. These issues included living in crowded dormitories, poor mask compliance at residential areas, conditions at workplaces. Stakeholders also noted issues pertaining to family attachment and non-compliance with quarantine facilities. Another common theme reported by migrant workers and healthcare providers was barriers related to non-vaccine supply chain management. Myanmar migrant workers also identified complacency as an issue. For example, one participant reported that migrant workers tend to

**Table 1. Background characteristics of study participants.**

| Type | Age | Sex | Occupation |
|---|---|---|---|
| Myanmar migrant worker 1 | 42 | Male | Factory worker |
| Myanmar migrant worker 2 | 43 | Male | Construction worker |
| Myanmar migrant worker 3 | 27 | Male | Factory worker |
| Myanmar migrant worker 4 | 28 | Male | Factory worker |
| Myanmar migrant worker 5 | 39 | Male | Seafarer |
| Myanmar migrant worker 6 | 28 | Male | Seafarer |
| Myanmar migrant worker 7 | 34 | Female | Construction |
| Thai employer 1 | Unspecified | Male | Construction supervisor |
| Thai employer 2 | Unspecified | Male | Construction supervisor |
| Thai employer 3 | Unspecified | Male | Factory manager |
| Thai employer 4 | Unspecified | Female | Factory Human Resources officer |
| Thai employer 5 | Unspecified | Female | Factory safety officer |
| Thai employer 6 | Unspecified | Female | Factory Human Resources officer |
| Thai healthcare provider 1 | Unspecified | Female | Nurse |
| Thai healthcare provider 2 | Unspecified | Female | Nurse |
| Thai healthcare provider 3 | Unspecified | Female | Doctor |
| Thai healthcare provider 4 | Unspecified | Female | Nurse |
| Thai healthcare provider 5 | Unspecified | Male | Doctor |
| Thai healthcare provider 6 | Unspecified | Male | Doctor |
| Thai healthcare provider 7 | Unspecified | Female | Nurse |
| Thai healthcare provider 8 | Unspecified | Female | Nurse |
| Thai healthcare provider 9 | Unspecified | Female | Nurse |

remove their masks or ignore guidelines when supervisors were not present in their work-places. Participants also reported financial constraints in purchasing face masks. Stakeholders also identified lack of health education and promotion as barriers. One particularly common issue identified by Thai employers was language barrier and the lack of interpreters when communicating with the Myanmar migrants. Fig 1 illustrates these themes and sub-themes as a coding tree chart.

## Key themes related to the barriers of COVID-19 vaccination

Regarding COVID-19 vaccination, one barrier consistently reported by all three stakeholder groups was fear of vaccination (Table 3), including concerns for side effects, overall vaccine hesitancy, and unspecified fear. One migrant worker participant reported personal experience with adverse effects of vaccination and refusal of subsequent doses, whereas a Thai employer used a reminder of reduced vaccination support as a way to motivate vaccination. Another common theme was the need for health education and health promotion, including educational materials in a language they understand, which could help to encourage uptake of booster doses. All three stakeholder groups mentioned issues pertaining to vaccine supply chain management, including lack of vaccination opportunity, insufficient vaccine supply, logistical challenges, and delays in both procurement and delivery of vaccines. Migrant workers and Thai employers mentioned policy related barriers, such as the requirement for legal documents, the need for the Thai employers to cover the cost of vaccination, and the prioritization of Thai citizens over Myanmar migrants. Thai employers and healthcare providers also frequently reported language barriers, including in vaccination-related tasks such as documentation, vaccination data entry, providing vaccination history, and communicating

**Table 2. Summary of themes, sub-themes, and codes regarding barriers to COVID-19 protective behaviors adherence among migrant workers from the perspectives of Myanmar migrant workers, Thai employers, and Thai healthcare providers.**

| Theme | Sub-theme | Codes | Quote No. | Example quotes or excerpts |
|---|---|---|---|---|
| Migrant workers' Theme 1: **Complacency** | None | ■ Relaxation of the preventive measures<br>■ Lack of supervision, careless attitude, and insufficient health officials' engagement<br>■ Perceived reduced risk<br>■ Variable effectiveness of preventive measures and non-adherence without supervision<br>■ Conditional mask wearing | 138, 142, 146, 154, 162, 166, 210, 211 | *"The preventive measures were mostly effective, but there were some that were less effective...Some people tend to be careless. When the supervisors aren't around, they remove their masks or leave them hanging....When in the dining area, people are told not to talk or share food, but they do it anyway. They even share water bottles.... They look around to see if anyone is watching, then start eating together. If someone gets infected, the whole factory is at risk."*<br>[Myanmar Migrant No. 3, Quote No. 166] |
| Migrant workers' Theme 2: **Financial constraints** | Financial barrier | ■ Financial burden for buying masks | 140 | *"Yes, [buying masks] does [become a financial burden]...they are quite expensive."*<br>[Myanmar Migrant No. 2, Quote No. 140] |
| Migrant workers' Theme 3: **Health education and health promotion barrier** | Lack of health education | ■ Insufficient health officials' Engagement<br>■ Lack of educational sessions<br>■ Lack of information<br>■ Limited IEC materials<br>■ Limited knowledge about preventive measures | 147, 160, 170, 192, 193, 197, 199, 203, 212 | *"No, [health officials] don't [encourage the worker to get vaccinated or provide education about it]...Employers should provide more information and encourage workers to get vaccinated to prevent the spread of the virus."*<br>[Myanmar Migrant No. 2, Quote No. 160] |
| Migrant workers' Theme 4: **Health service delivery barrier** | Lack of access by healthcare workers | ■ Insufficient health officials' engagement<br>■ Lack of educational sessions and lack of screening | 150, 151, 205, 208 | *"Since we're out at sea, and we're not on an island, it's difficult to meet with Thai health workers."*<br>[Myanmar Migrant No. 5, Quote No. 205] |
| Migrant workers' Theme 5: **Lifestyle and habit related barrier** | Barrier related to living and working conditions | ■ Poor living conditions<br>■ Risky, shared, poor living conditions<br>■ Risky dormitory environment and poor living conditions<br>■ Poor living conditions, risky dormitory environment, shared facilities, and poor mask wearing habits<br>■ Risky workplace environment<br>■ Lack of eating place<br>■ External contact<br>■ Poor working conditions<br>■ Lack of hygiene support, poor working conditions, and lack of restriction<br>■ Difficulty in social distancing due to the nature of the work<br>■ Poor working conditions, lack of separate eating areas, difficulty maintaining social distancing, and poor enforcement mechanism | 128, 132, 144, 152, 163, 168, 171, 172, 185, 186, 191, 194, 195, 196, 198, 207, 216, 217, 218, 219 | *"There are no specific separate areas for eating places at the worksite. While eating, we almost sit together with other employees, like there are about 6-7 people who usually eat with us. We are divided into several small groups while eating....The major one is that it is difficult to maintain distancing measures during working. [The employers] don't really restrict it. Our work is difficult to follow distancing measures."*<br>[Myanmar Migrant No. 7, Quote No. 216]<br>*"It's quite difficult for us to maintain social distancing in our residence because the rooms are small and close to each other. The ventilation is somewhat inadequate, and the room space is quite limited."*<br>[Myanmar Migrant No. 7, Quote No. 219] |
| Migrant workers' Theme 6: **Non-vaccine supply chain management barrier** | Lack of supplies | ■ Lack of masks<br>■ Limited supply of mask<br>■ Lack of detergents<br>■ Lack of soaps<br>■ Lack of disinfectants<br>■ Lack of hand sanitizers, masks, temperature checks, and inability to make social distancing | 136, 137, 139, 141, 143, 149, 155, 161, 182 | *"In the production line, we don't have disinfectant, but I sanitize after using the bathroom. During the peak of COVID, they came and sprayed us individually."*<br>[Myanmar Migrant No. 4, Quote No. 182] |
| Thai employers' Theme 1: **Lifestyle and habit related barrier** | Barrier related to living and working conditions | ■ Employer could not manage unauthorized entry by vendors<br>■ Close contact due to the nature of the work | 43, 44 | *"We told [migrant workers] not to leave unless necessary and kept vendors out. Despite our efforts, a vendor managed to get in and brought COVID-19. This caused the whole camp to get COVID"*<br>[Thai employer No. 5, Quote No. 43] |
| | Family attachment | ■ Family attachment | 1, 45 | *"Some people have children, and mothers and children did not wish to be separated…If they were separated, there would be an issue of [children] being unable to stay with their father and wants to stay with their mother. Some people allowed children to stay with their moms."*<br>[Thai employer No. 1, Quote No. 1] |

*(Continued)*

**Table 2.** (Continued)

| Theme | Sub-theme | Codes | Quote No. | Example quotes or excerpts |
|---|---|---|---|---|
| Thai employers' Theme 2: **Language barrier** | None | ■ Language barrier<br>■ Lack of interpreter or translator | 27, 28 | *"The language issue was, like, there were [migrants] who didn't understand. They couldn't speak Thai. And it's like, why do we have to do this every day? It was a little difficult to talk about that matter."*<br>[Thai employer No. 2, Quote No. 27] |
| Thai healthcare providers' Theme 1: **Fear** | Fear of medical personnel | ■ Non-disclosure of reporting of covid infections (Transparency) | 101 | *"I think [migrant workers] would want to keep [the infection] a secret. Those who are sick want to keep it a secret because they want to keep working…They are quite afraid of medical personnel and having COVID-19 makes them feel very bad. So even the patients themselves might want to keep it a secret, does not want to know or acknowledge that they are infected."*<br>[Thai Healthcare provider No.5, Quote No.101] |
| Thai healthcare providers' Theme 2: **Lifestyle and habit related barrier** | Non-compliance to quarantine | ■ Irrelevance or resistance to Quarantine | 99 | *"I don't have direct information on [migrants' behavior, like following social distancing measures]. However, I've heard from the team that initially, when we separated high-risk groups from non-infected and infected groups, there were some who didn't see the point in isolation because they felt fine and resented being quarantined for 14 days without symptoms. Yes, in the first wave, they had to be quarantined for 28 days in total, which included 14 days of treatment and 14 days of isolation afterward. This extended period caused significant stress and loss of income."*<br>[Thai Healthcare provider No. 5, Quote No.99] |
| Thai healthcare providers' Theme 3: **Non-compliance by migrants or employers** | Non-disclosure of outbreak occurrence | ■ Non-disclosure of reporting of covid infections (Transparency) | 100 | *"There was also a period when there was an outbreak in a factory, but the factory kept it a secret, which delayed treatment and caused the outbreak to spread widely.*<br>[Thai Healthcare provider No.5, Quote No. 100] |
| Thai healthcare providers' Theme 4: **Non-vaccine supply chain management barrier** | Lack of supplies | ■ Insufficient supply of hygiene items | 102 | *"COVID-19 preventive measures seem to have relaxed somewhat [in factories]. For example, there used to be numerous alcohol gels, but now there are fewer. Hand soap in the bathrooms is not always available, unlike before."*<br>[Thai Healthcare provider No.5, Quote No. 102] |

about medical histories. Migrant workers and healthcare providers also reported data system barriers, including the need for identification and documents for both vaccination data and registration system, service delays, data entry problem, communication, and coordination challenges. We also presented these themes and sub-themes in Fig 2 as a coding tree chart.

Analysis of the frequency in which themes appeared for both adherence to COVID-19 prevention behaviors and COVID-19 vaccination showed that the theme that appeared most frequently for adherence to COVID-19 protective measures was lifestyle and habit-related barriers (Supplementary Table 1). On the other hand, the most common theme for COVID-19 vaccination was vaccine supply chain management (Supplementary Table 2).

## Discussion

Our study examines the barriers to adherence to COVID-19 protective behaviors and COVID-19 vaccination among Myanmar migrant workers in Southern Thailand during the COVID-19 pandemic according to the workers, their Thai employers, and healthcare providers. Each of the three stakeholder groups revealed several key themes related to barriers affecting COVID-19 related health behaviors among migrants. Two common themes emerged regarding adherence to COVID-19 protective behaviors: lifestyle and habit-related barriers, and non-vaccine supply chain management barrier. For COVID-19 vaccination, three

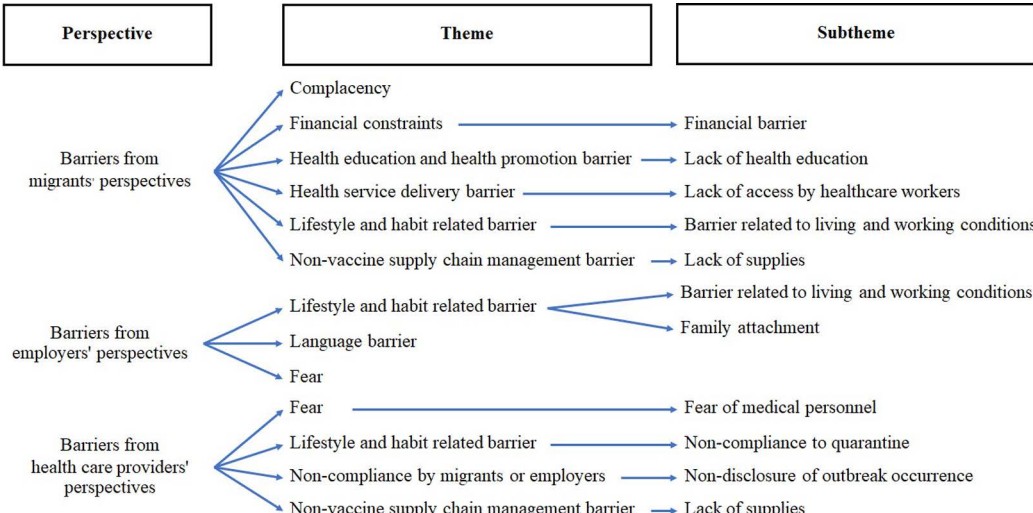

**Fig 1. Coding tree for barriers related to adherence to COVID-19 protective behaviors at workplace and residence.**

common themes were identified: fear, barriers related to health education and health promotion, and vaccine supply chain management.

We found that lifestyle and habit-related barriers hindered adherence to COVID-19 preventive measures, similar to the findings of previous qualitative studies conducted among migrant workers in Qatar [41], Norway [42], and Uganda [43]. The findings also concurred with those of previous reports conducted in Thailand during the COVID-19 lockdown among factory workers [16], construction workers [44], and fishery workers [45]. In Thailand, Myanmar migrant workers tend to live in overcrowded accommodations with poor hygienic conditions [45–49], which can increase the risk of further outbreaks of COVID-19 and other communicable diseases [14,16]. The improvement of migrants' living and working conditions may require shifting of existing economic equilibrium between conditions and costs on the part of the migrants and their employers [2]. Thus, future studies should consider exploring approaches to shift this equilibrium in a direction that enables effective disease prevention and control.

Non-vaccine supply chain management issues included the shortage of essential hygiene items and personal protective equipment (PPE), which could have improved the overall adherence to COVID-19 and other respiratory infection preventive measures [50]. In Malaysia, one study among Myanmar refugees and irregular migrants in Malaysia reported lack of employer support for masks and hand sanitizers during COVID-19 pandemic [51]. In Thailand, although over 200,000 migrants in also requested hygiene items and food from migrant support groups [19], survey data from the International Organization of Migration reported that the migrants faced many challenges including an inadequate supply of essentials [52]. Future studies should explore the determinants of supply inadequacy to make evidence-based recommendations to stakeholders to prepare for future outbreaks and epidemics.

Regarding barriers to COVID-19 vaccination, our participants reported the lack of health education and health promotion as barriers. Previous studies also made similar reports, i.e., that the lacks of trustworthy information and culturally-accessible information deterred ethnic minorities from receiving the COVID-19 vaccinations [53,54]. Our participants also reported that they feared COVID-19 vaccination, particularly about the vaccine's side effects. These findings were similar to previous studies, which reported that the reasons for

**Table 3. Summary of themes, sub-themes, and codes regarding barriers to COVID-19 vaccination among migrant workers from the perspectives of Myanmar migrant workers, Thai employers, and Thai healthcare providers.**

| Theme | Sub-theme | Codes | Reference | Example quotes or excerpts |
|---|---|---|---|---|
| Migrant workers' Theme 1: **Data system barrier** | Vaccination data system | ■ Documentation requirement | 189 | *"Some [migrant workers] miss getting vaccinated because they didn't know what documents were needed?"* [Myanmar migrant No. 4, Quote No. 189] |
| Migrant workers' Theme 2: **Fear** | Fear of vaccination | ■ Concern about vaccine side effects<br>■ Negative perception about vaccines<br>■ Experience of side effects and concern about alternative preventive measure | 175, 177, 179, 187, 188, 221, 222 | *"When I received the initial dose, I experienced fever, weakness, and nausea, which lasted for nearly five days, preventing me from going to work. So, for the second dose, I refused to accept it personally. But I also asked my husband's suggestion, and he agreed not to accept it. And I think I can wear masks that can also prevent covid transmission to others."* [Myanmar migrant No. 7, Quote No. 221] |
| Migrant workers' Theme 3: **Financial constraint** | Vaccine-related financial barrier | ■ Vaccination cost and out-of-pocket payment for vaccine | 180 | *"Not at our factory, but I've heard about it in other places. Some workers had to pay for their own vaccinations. Some workers didn't get vaccinated at all."* [Myanmar migrant No. 3, Quote No. 180] |
| Migrant workers' Theme 4: **Vaccine supply chain management** | Lack of opportunity for vaccination | ■ Lack of notification for vaccination<br>■ Long waiting to get vaccination or delay in vaccination | 157, 190 | *"Oh, our workplace had a Covid outbreak after we got vaccinated. Some migrants hadn't been vaccinated yet, at that time. They called for vaccinations after, and not everyone was called at the same time."* [Myanmar migrant No. 5, Quote No. 190] |
| Migrant workers' Theme 5: **Health education and health promotion** | Lack of health education | ■ Lack of IEC materials for vaccination | 156 | *" [Our employer] didn't distribute educational materials about COVID-19. No posters, in Burmese language, either. No signs for hand washing too…"* [Myanmar migrant No. 2, Quote No. 156] |
| | Lack of information about vaccination | ■ Lack of information about vaccines<br>■ Lack of information about vaccine and poor knowledge about the vaccine | 159, 178 209 | *"I haven't heard much about side effects of COVID-19 vaccine being severe. [Our manager] didn't discuss about the vaccines with us before we got vaccinated. They just encouraged us to get vaccinated. If workers refused, they still had to get vaccinated."* [Myanmar migrant No. 3, Quote No. 178] |
| Migrant workers' Theme 6: **Policy barrier** | Lack of health coverage | ■ Documentation requirements | 158 | *"I think it will be difficult to get vaccinated without a work permit There are people who couldn't get the vaccine because they just arrived, and their documents are not complete. They're not listed in the system here. They're from Myanmar. They from your dormitory, but they have moved out now."* [Myanmar migrant No. 2, Quote No. 158] |
| Thai employers' Theme 1: **Fear** | Fear of vaccination | ■ Negative perceptions of vaccination<br>■ Concern of vaccine side effects<br>■ Fear of vaccination<br>■ Fear of vaccination and responsibility<br>■ Reluctance to Vaccinate<br>■ Vaccine hesitancy | 3, 4, 13, 31, 38, 61 | *"Initially, there was fear of vaccination among migrants. We told them if they didn't get vaccinated now, they would have to handle it themselves later. This made them more willing to get vaccinated."* [Thai employer No. 3, Quote No. 38] |
| Thai employers' Theme 2: **Health education and health promotion** | Lack of information about vaccination | ■ Fear of vaccination | 29 | *"Well, [the migrant workers] had fears, because they didn't understand why they needed the shots. And in the beginning, foreigners were not.... what do you call this.... they were the last group to receive vaccination. So, they had a hard time understanding why they didn't get the vaccine... After we opened and allowed them to get vaccinated, they cooperated well."* [Thai employer No. 2, Quote No. 29] |
| Thai employers' Theme 3: **Policy barrier** | Lack of health coverage | ■ Vaccination cost | 37 | *"Initially, the company paid about 1,000 baht per person through the Industrial Council. Later, it wasn't necessary."* [Thai employer No. 3, Quote No. 37] |
| | Vaccination policy | ■ Frustration due to being not prioritized for vaccine | 47 | *"When the factory started operating, there was frustration among workers. They wondered why their needs weren't prioritized for vaccine."* [Thai employer No. 5, Quote No. 47] |

*(Continued)*

**Table 3.** (Continued)

| Theme | Sub-theme | Codes | Reference | Example quotes or excerpts |
|---|---|---|---|---|
| Thai employers' Theme 4: **Language barrier** | None | ■ Language barrier<br>■ Lack of interpreter or translator | 15, 26, 41 | *"We didn't have a dedicated interpreter but had Burmese employees who could translate. They helped communicate information during the pandemic"*<br>[Thai employer No. 4, Quote No. 41] |
| Thai employers' Theme 5: **Vaccine supply chain management** | Lack of vaccine supply for migrants | ■ Limited availability of vaccine<br>■ Need for private purchase of vaccine<br>■ Lack of vaccine opportunity<br>■ Fear of vaccination<br>■ Lack of health education combined with fear and inequity<br>■ Inadequate supply of the vaccine to meet the demand<br>■ Delay in vaccination<br>■ High demand of vaccine | 8, 10, 11, 34, 36, 39, 49, 53, 54, 56, 57, 58, 59 | *"In the beginning of vaccine roll-out, there was no vaccine for Burmese migrants. It was available only to Thai people living in the area...so some people got the vaccine, while some didn't even get them. Myself, the safety officer, and my boss all had to follow the news about where vaccinations are available, and we had to check that this was available to the general public and not just local residents. We had to follow the news closely. My boss even had to make purchase for the two doses. It was Sinopharm. Because there was no vaccine for Burmese people."*<br>[Thai employer No. 1, Quote No. 8] |
| | Inability to make reservation for vaccination | ■ Lack of awareness of vaccine reservation | 9 | *"We had to buy the COVID-19 vaccine at Sinopharm at that time. Ah, we have to make a reservation. My boss doesn't know how to find a way to make a reservation in time. He found every way to make reservations in next time."*<br>[Thai employer No. 1, Quote No. 9] |
| | Delay in vaccine procurement | ■ Slow pace of the government's vaccine procurement effort<br>■ Limited availability of vaccine<br>■ Delay in vaccination<br>■ Slow in government's vaccine procurement | 40, 50, 51, 52, 57 | *"Initially, vaccine procurement was slow due to government policies. Screening was challenging without ATK tests. Once we had ATKs, we could quickly identify and isolate cases within 10 minutes, unlike waiting two days for RT-PCR results. Early challenges included delays in equipment and vaccines, but once we had the necessary resources, our response improved significantly."*<br>[Thai employer No. 6, Quote No. 57] |
| | Delay in vaccine delivery | ■ Delay in vaccination | 48, 60 | *"There was dissatisfaction among the migrant workers about not being tested quickly enough.*<br>*They were eager to get vaccinated and complained about the delays."*<br>[Thai employer No. 5, Quote No. 48] |
| Thai healthcare providers' Theme 1: **Data system barrier** | Barrier related to vaccination data system | ■ Problem in planning for vaccination<br>■ Problem in data entry for vaccination<br>■ No central point of contact"<br>■ Problem in data entry for vaccination (app-related problem)<br>■ Documentation requirement<br>■ Requirement of identification number<br>■ Problem in individual identification<br>■ Documentation requirement and problem in data entry for vaccination<br>■ Language barrier and problem in data entry for vaccination<br>■ Service delay due to problem in vaccine registration system<br>■ Vaccination database management issue and the need for ongoing promotion for vaccination | 65, 66, 67, 68, 77, 78, 79, 85, 91, 92, 95, 104, 113, 116 | *"The main weakness providing vaccines to foreign workers is communication. Some foreign workers received the Sinopharm vaccine from private hospitals, such as ((redacted)) Hospital or other hospitals, not just in Hat Yai. Sometimes, other district hospitals also provide services because businesses can pay for services per case, such as vaccination fees and equipment costs. When these foreign workers come to us for their third dose, we sometimes cannot find their first and second dose records in the system. We then must ask the businesses to coordinate and obtain information from the hospitals that administered the previous doses. Sometimes, this information is not available, so we must record the third dose as their first dose."*<br>[Thai healthcare provider No. 4, Quote No. 95] |
| | Vaccination registration problems | ■ Delay in customer bookings for vaccination<br>■ Delay in vaccination delivery<br>■ Problem in data entry for vaccination and documentation requirement | 73, 81, 84 | *"The company-arranged vaccinations were generally smooth because they handled procurement and logistics. The problem arose with ((redacted)) vaccines due to delays and issues with customer bookings. People would get impatient and OPD for government vaccines, causing confusion and complaints about refunds."*<br>[Thai healthcare provider No. 4, Quote No.73] |
| Thai healthcare providers' Theme 2: **Fear** | Fear of vaccination | ■ Fear of vaccination | 70, 119, 124 | *"I remember there was a period when Thai people had a lot of issues and concerns about the vaccine. I wondered if the Burmese workers had similar concerns that caused disruptions. They might have had some fear or anxiety internally for vaccination."*<br>[Thai healthcare provider No. 7, Quote No.119] |

*(Continued)*

**Table 3.** (Continued)

| Theme | Sub-theme | Codes | Reference | Example quotes or excerpts |
|---|---|---|---|---|
| Thai health-care providers' Theme 3: **Health education and health promotion** | Lack of information about vaccination | ■ Low booster uptake | 106 | *"Now, we have enough vaccines, even excess. Few [migrants] come for booster shots. The current challenge is encouraging Burmese workers to get vaccinated and raising awareness about booster shots. They can get vaccinated at hospitals or health centers."*<br>[Thai healthcare provider No. 5, Quote No.106] |
| | Lack of health promotion | ■ Communication and Compliance Strategy | 115 | *"Currently, we have enough vaccines. Hardly anyone refuses the third dose or booster shots. What we need is continuous monitoring of businesses or employers to ensure they are vaccinated. We need support in disseminating information about where workers can get vaccinated and raising awareness about the importance of vaccination. We would like intermediaries to help communicate and publicize the importance of vaccination to workers and employers."*<br>[Thai healthcare provider No. 6, Quote No.115] |
| Thai health-care providers' Theme 4: **Health service delivery** | Lack of access by healthcare workers | ■ Difficult access to the factory | 97 | *"Sometimes, if the factory doesn't need us, it's very difficult for us to reach them. Entering a factory can be quite challenging for us."*<br>[Thai healthcare provider No. 5, Quote No. 97] |
| Thai health-care providers' Theme 5: **Language barrier** | None | ■ Language barrier<br>■ Documentation requirement<br>■ Rely on interpret or translator<br>■ Language barrier and problem in data entry for vaccination<br>■ Vaccine history issue and language barrier<br>■ Language barrier and the lack of interpreter/translator<br>■ Language barrier and vaccine availability | 69, 80, 83, 86, 87, 93, 94, 96, 107, 112, 114, 118, 120, 123, 125, 126 | *"For Burmese migrant workers, communication is the biggest challenge, like taking medical histories where some important information might get lost. With Thai people, they might be more particular or demanding, asking things like if they can get vaccinated here or if they can have half a dose."*<br>[Thai healthcare provider No. 9, Quote No.125] |
| Thai health-care providers' Theme 6: **Vaccine supply chain management** | Delay in vaccine procurement | ■ Delay in vaccination | 62 | *"People could reserve their doses, and it was handled collectively through private associations, which then managed the bulk orders. So, for the government vaccines, it depended on their policies—who to vaccinate first and with which vaccine. Private vaccines had to be ordered in batches, which sometimes arrived later than expected. Initially, we used government-provided vaccines according to their policies."*<br>[Thai healthcare provider No. 1, Quote No. 62] |
| | Lack of vaccine supply for migrants | ■ Requirement of identification number<br>■ Limited availability of vaccine<br>■ New procedure for vaccination and limited vaccine availability<br>■ Postponing vaccination schedule | 64, 74, 76, 82, 88, 89, 90, 103, 105, 121, 122 | *"In the initial phase of vaccination, the government required information to be entered using the 13-digit ID card number. Therefore, in the beginning, those without ID cards couldn't get vaccinated because we had to input the 13-digit number. Even if they were Thai, without the number, we couldn't vaccinate them. This was [according to the] regulation. So, initially, foreign workers were not included at all. But later, the government adjusted the system to include foreign workers who had their own identification numbers."*<br>[Thai healthcare provider No. 1, Quote No. 64] |
| | Vaccine supply and demand | ■ Problem in managing vaccine supply | 75 | *"Actually, there were rules on vaccination, but the rules have stopped. The company probably knew about this. If the number exceeded 10 person, and all of a sudden we could vaccine 11 persons, problems would arise at the 11th person and there would be complaints... in the past, we would have the last vial open, and only 5 people showed up, the other 5 became our responsibility to make calls for people to get as many of their family members and staff to come for vaccination. ..... Things later improved. They gave us the green light and only vaccinate until the end of the vaccination time."*<br>[Thai healthcare provider No. 1, Quote No. 75] |

*(Continued)*

**Table 3.** (Continued)

| Theme | Sub-theme | Codes | Reference | Example quotes or excerpts |
|---|---|---|---|---|
| | Logistical constraints | ■ Capacity challenge<br>■ Accessibility to vaccination site | 110, 111 | *"There is the issue of whether migrant workers get permission from their employers to take time off work and if the employers can organize their schedules because they still have work responsibilities. So, the first issue is whether they have the opportunity to reach the vaccination point."*<br>[Thai healthcare provider No. 6, Quote No. 111] |

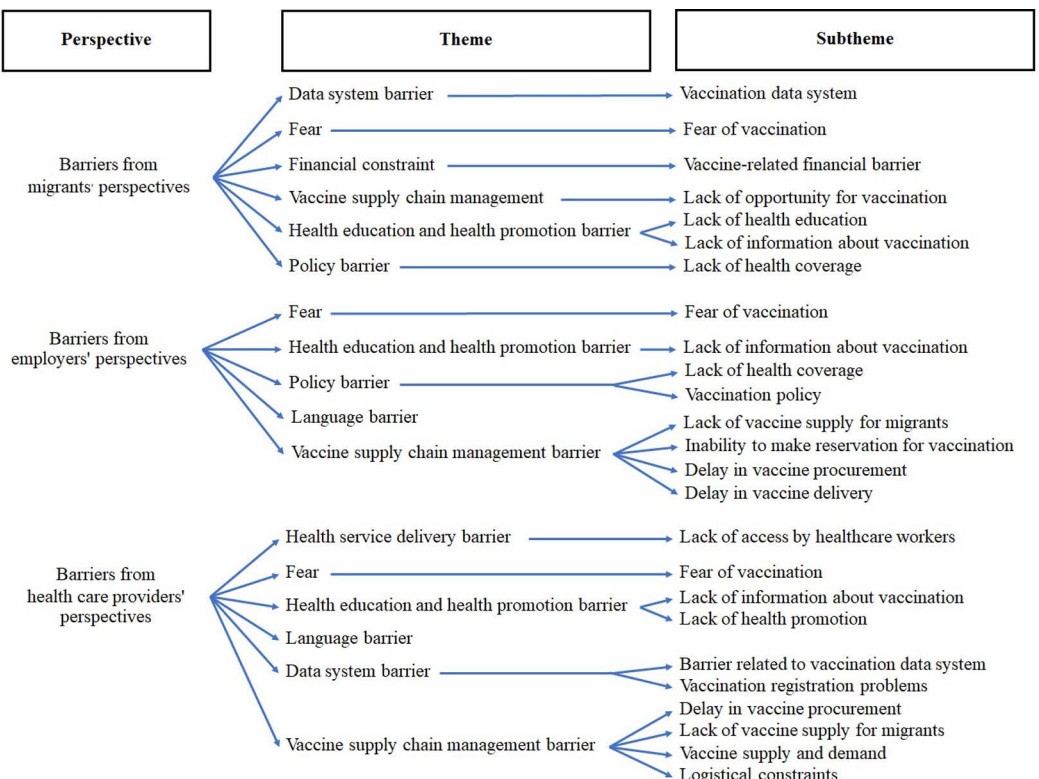

**Fig 2. Coding tree for barriers related to COVID-19 vaccination among migrant workers from various perspectives.**

COVID-19 vaccine uptake (or lack thereof) among minority groups and migrants included the fear of needles and side effects of vaccines [55,56], safety concerns [57,58]. A negative attitude toward the vaccine could be partly attributed to the influence of information sources [53]. The lack of documentation of less common side effects of vaccines also could have contributed to the mistrust [59]. These findings highlight that addressing fears might be useful in increasing vaccination uptake. Future studies should explore the determinant of fears in a holistic manner, such as considering the influence of financial impact of vaccination as a potential determinant [60]. Our study only included migrants with relatively low socioeconomic status. Studies have found that context of migration [61], language barrier and employer support [51] are associated with COVID-19 vaccine intention. These attributes tend to correlate with socioeconomic status; thus it is possible that skilled Myanmar migrants working in professional services might have different levels and domains of vaccination concerns compared to our participants. Future studies should consider expanding the scope of the study to include migrants in higher socioeconomic status.

Vaccine supply chain management barriers in our study included the lack of vaccination opportunity, insufficient vaccine supply, delays in vaccine procurement and delivery, and logistical challenges. Other studies also suggested that logistical issues have driven low COVID-19 vaccine uptake among migrants and racial minorities [53,57]. Disparities in vaccine delivery have been evident, particularly affecting minority and vulnerable communities, such as undocumented migrants in Thailand and the Roma community in Europe [62]. However, in addition to logistics, we also found structural issues such as difficulties in arranging vaccination appointments, similar to the findings of a previous study [54]. Despite the allocation of 500,000 doses of the COVID-19 vaccine for migrant workers to combat labor shortages and promote economic recovery [63], migrant workers who were insured by the Social Security Scheme (SSS) were prioritized over uninsured migrants [64], or the number of doses received by migrant workers might be limited [16,65]. The situation seemed to have improved by late 2021 [16]. However, in that year, approximately 1.3 million uninsured migrants did not receive any vaccinations in Thailand [66]. This highlights the critical need for more equitable distribution systems that ensure vaccine access regardless of insurance status. To improve vaccine access for migrant populations, supply chain management should be optimized through stratified budgeting, buffer stock systems, and mobile vaccination units, while strengthening cold chain infrastructure.

Our study also identified policy-related vaccination barriers, specifically documentation requirements, vaccination cost, and vaccine policy. Although the Thai government implemented an inclusive COVID-19 policy with support from the Ministry of Public Health and the Thai Red Cross [35], irregular Myanmar migrants still faced documentation issues as barriers to vaccination, whereas legalization may entail significant costs [67]. In addition to policy, there were barriers within the healthcare system including the necessity for identification in various vaccination procedures as well as data entry problems. Myanmar irregular migrants might be unfamiliar with the COVID-19 vaccination registration system [67]. Existing mobile-phone based applications, such as MorPhrom, can track vaccination records but do not fully support undocumented migrants, creating significant employment obstacles as many employers require proof of vaccination [35]. Therefore, partnerships with civil society organizations and employers, alongside policy adjustments that include amnesty provisions for undocumented migrants and removal of deterrent documentation requirements, ultimately creating an equitable and efficient vaccination system for all migrants regardless of their status. Moreover, structural barriers should be addressed by simplifying registration processes, removing insurance-based prioritization, and establishing multiple convenient vaccination points with flexible scheduling to accommodate work schedules.

In addition to health system and health policy, language is another commonly identified barrier to COVID-19 vaccination. This particular finding concurred with previous qualitative studies that described the difficulties of migrant from Myanmar in understanding vaccination procedures [67] and healthcare information [20]. Language is a common barrier to healthcare access for migrants in Thailand [68] and elsewhere [69]. Migrant workers from Myanmar typically have a limited command of Thai [70,71]. As most foreigners would need approximately 100 hours to learn Thai to sufficiently communicate [72,73], our study findings suggest that stakeholders in migrant health should identify potential translators and interpreters in contingency plans for future outbreaks and pandemics.

### Strengths and limitations of the study

The strength of this study was the use of the native languages of the participants (Burmese and Thai) during interviews, which precluded potential information bias from interpretation or translation errors. However, a number of limitations should be considered in the interpretation

of our study findings. Firstly, although our protocols did not exclude undocumented migrants, we were only able to recruit documented migrants working in manual labor occupations common in southern Thailand. These circumstances limited the generalizability of our study findings. The barriers among undocumented migrants might have been vastly different from those reported by our participants. Secondly, despite the use of primary languages to interview the participants, the possibility of social desirability bias could not be precluded from the study findings considering the sensitive nature of some of the interview questions. Thirdly, in our study, the transcription and translation of the interviews were different for those of the Myanmar migrants and those of the Thai employers and healthcare workers. The choices were based on the subjective judgment of the investigators. Such practice might have introduced potential inaccuracies. However, considering that we used verbatim texts for our analyses, we anticipate these differences to be small, particularly in the context of the broader study findings. Future studies should consider adapting methods to overcome these limitations.

## Conclusion

In this qualitative study, we identified barriers to COVID-19 protective behaviors and vaccination among Myanmar migrants in Thailand. Supply chain management was a common theme in both barriers to vaccination and barriers to adherence to COVID-19 preventive measures. Moreover, each domain also had additional themes, such as lifestyle and habit-related barriers, lack of health education and health promotion, and fear of vaccination. These findings provide potentially useful insights for policymakers and healthcare providers working to enhance vaccine equity among migrant populations and prepare an inclusive plan to respond to future health emergencies. However, limitations regarding the lack of generalizability and social desirability bias should be considered in the interpretation of our study findings. Future research should focus on evaluating targeted interventions that address both systemic barriers and individual concerns, particularly in contexts where migrants face multiple vulnerabilities.

## Supporting Information

**S1 Table. Frequency of themes during interviews about barriers to adherence to COVID-19 protective measures**
(DOCX)

**S2 Table. Frequency of themes during interviews about barriers for COVID-19 vaccination**
(DOCX)

**S3 Table. COREQ (COnsolidated criteria for REporting Qualitative research) Checklist**
(DOCX)

**S1 Data. Data File with Redacted Names**
(XLSX)

## Acknowledgments

This study was conducted as part of the first author's doctoral thesis in the Department of Epidemiology, Prince of Songkla University, Thailand. We wish to thank all migrant workers, authorized persons from hospitals, factories, construction and fishery sites who actively and voluntarily participated in this study. We would like to acknowledge all local key informants who helped with data collection with immense support and cooperation.

## Author contributions

**Conceptualization:** Hein Htet, Hutcha Sriplung, Virasakdi Chongsuvivatwong.

**Data curation:** Hein Htet, Wit Wichaidit.

**Formal analysis:** Hein Htet, Wit Wichaidit.

**Funding acquisition:** Virasakdi Chongsuvivatwong.

**Investigation:** Hein Htet, Aungkana Chuaychai, Tiida Sottiyotin, Kyaw Ko Ko Htet.

**Methodology:** Hein Htet, Hutcha Sriplung, Virasakdi Chongsuvivatwong.

**Project administration:** Hein Htet, Aungkana Chuaychai, Tiida Sottiyotin, Kyaw Ko Ko Htet.

**Resources:** Virasakdi Chongsuvivatwong.

**Software:** Hein Htet, Wit Wichaidit.

**Supervision:** Virasakdi Chongsuvivatwong.

**Validation:** Hein Htet, Virasakdi Chongsuvivatwong.

**Visualization:** Hein Htet, Wit Wichaidit.

**Writing – original draft:** Hein Htet, Wit Wichaidit.

**Writing – review & editing:** Hein Htet, Wit Wichaidit, Virasakdi Chongsuvivatwong.

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
