## [Decision Letter · Decision Letter 0]

14 Oct 2024

Dear Dr. Wichaidit,

Thank you for submitting your manuscript to PLOS ONE. After careful consideration, we feel that it has merit but does not fully meet PLOS ONE’s publication criteria as it currently stands. Therefore, we invite you to submit a revised version of the manuscript that addresses the points raised during the review process.

We look forward to receiving your revised manuscript.

Kind regards,

Kyaw Lwin Show, MPH, PhD

Academic Editor

PLOS ONE

4. We note that this data set consists of interview transcripts. Can you please confirm that all participants gave consent for interview transcript to be published?

If they DID provide consent for these transcripts to be published, please also confirm that the transcripts do not contain any potentially identifying information (or let us know if the participants consented to having their personal details published and made publicly available). We consider the following details to be identifying information:

- Names, nicknames, and initials

- Age more specific than round numbers

- GPS coordinates, physical addresses, IP addresses, email addresses

- Information in small sample sizes (e.g. 40 students from X class in X year at X university)

- Specific dates (e.g. visit dates, interview dates)

- ID numbers

Or, if the participants DID NOT provide consent for these transcripts to be published:

- Provide a de-identified version of the data or excerpts of interview responses

- Provide information regarding how these transcripts can be accessed by researchers who meet the criteria for access to confidential data, including:

a) the grounds for restriction

b) the name of the ethics committee, Institutional Review Board, or third-party organization that is imposing sharing restrictions on the data

c) a non-author, institutional point of contact that is able to field data access queries, in the interest of maintaining long-term data accessibility.

d) Any relevant data set names, URLs, DOIs, etc. that an independent researcher would need in order to request your minimal data set.

For further information on sharing data that contains sensitive participant information, please see: https://journals.plos.org/plosone/s/data-availability#loc-human-research-participant-data-and-other-sensitive-data

If there are ethical, legal, or third-party restrictions upon your dataset, you must provide all of the following details (https://journals.plos.org/plosone/s/data-availability#loc-acceptable-data-access-restrictions):

1. A complete description of the dataset

2. The nature of the restrictions upon the data (ethical, legal, or owned by a third party) and the reasoning behind them

3. The full name of the body imposing the restrictions upon your dataset (ethics committee, institution, data access committee, etc)

4. If the data are owned by a third party, confirmation of whether the authors received any special privileges in accessing the data that other researchers would not have

5. Direct, non-author contact information (preferably email) for the body imposing the restrictions upon the data, to which data access requests can be sent

Additional Editor Comments:

Use the COREQ checklist for qualitative interviews and attach it as a supplementary document.Introduction: Information regarding the importance of adhering to COVID-19 protective behaviors and vaccination is missing.It would be better to explain the COVID-19 waves and the epidemic situation in Thailand for clearer understanding to the readers. Was data collection conducted during the peak of the epidemic, or before or after a major wave?Were the migrant workers you collected data from legal or undocumented workers? Legal and undocumented migrant workers may face different barriers.I am curious about the accuracy of Google Translate in translating Burmese and Thai texts into English. Did the study team review and correct the translations after using the tool?Provide the background characteristics of the participants for all three categories.Currently, the investigators have analyzed the information based on the type of participant. Please consider analyzing the data by theme, covering all stakeholders. For example, under the theme of health education barrier, presenting perspectives from migrant workers, employers, and healthcare providers would make the findings easier to understand.Move Tables 3 and 4 to the supplementary section

Reviewers' comments:

Reviewer's Responses to Questions

**Comments to the Author**

1. Is the manuscript technically sound, and do the data support the conclusions?

Reviewer #1: Yes

Reviewer #2: Partly

2. Has the statistical analysis been performed appropriately and rigorously?

Reviewer #1: N/A

Reviewer #2: No

3. Have the authors made all data underlying the findings in their manuscript fully available?

Reviewer #1: Yes

Reviewer #2: Yes

4. Is the manuscript presented in an intelligible fashion and written in standard English?

Reviewer #1: Yes

Reviewer #2: Yes

Reviewer #1: Thank you very much for your efforts in adding new information on perspectives regarding COVID-19 protective behaviours and vaccination among Myanmar migrant workers, Thai health care providers and Thai employers.

In the attachment, I included my recommendations.

Reviewer #2: The article focuses on research concerning Myanmar migrant workers; however, it lacks clarity regarding the specific group of workers being studied, whether regular or irregular migrants. It is crucial to differentiate between these groups as their experiences vary significantly. For instance, regular migrant workers typically have access to healthcare, whereas irregular migrant workers face restrictions due to their immigration status.

To address the research gap, a more comprehensive literature review is recommended to enhance the paper. For instance, the article's sample should encompass various groups such as migrant workers, meatworkers, NGOs, and local individuals.

• 'I Doubt Myself and Am Losing Everything I Have since COVID Came'-A Case Study of Mental Health and Coping Strategies among Undocumented Myanmar Migrant Workers in Thailand https://doi.org/10.3390/ijerph192215022

Regarding methodology, the study should provide detailed information on how participants were recruited. Additionally, the low number of migrant workers in the study raises concerns, especially considering that the data collection period occurred after the easing of COVID-19 restrictions in Thailand. The claim that data collection was impeded by COVID-19 restrictions should be verified.

Furthermore, the study should clarify how the data were coded and analyzed, whether through NVivo software or manual coding. Given that the study is purely qualitative, the presentation of findings should follow with qualitative research standards, favoring textual descriptions over tabular formats.

Authors are encouraged to devise a more effective chart or diagram to present the findings and integrate direct quotations alongside their interpretations. The discussion section should incorporate relevant studies from ASEAN countries,

• Preventive knowledge, attitude, and vaccination challenges for COVID-19 among Myanmar refugees and irregular migrants in Malaysia, https://doi.org/10.1016/j.jvacx.2023.100360

• Global Disparities in COVID-19 Vaccine Distribution: A Call for More Integrated Approaches to Address Inequities in Emerging Health Challenges https://doi.org/10.3390/challe14040045

Additionally, the study should offer practical policy implications, outline its strengths and limitations using sub-heading.

Provide a more detailed and impactful conclusion to enhance its scholarly contribution.

**Do you want your identity to be public for this peer review?** For information about this choice, including consent withdrawal, please see our Privacy Policy

Reviewer #1: No

Reviewer #2: No

---

## [Author Response · Author response to Decision Letter 1]

15 Dec 2024

RESPONSE TO THE EDITOR AND REVIEWERS' COMMENTS

Editor's comment

EDITOR’S COMMENT

1. We note that the grant information you provided in the ‘Funding Information’ and ‘Financial Disclosure’ sections do not match. When you resubmit, please ensure that you provide the correct grant numbers for the awards you received for your study in the ‘Funding Information’ section.

RESPONSE

We thank the reviewer for the comment, and we have checked the grant numbers accordingly.

EDITOR’S COMMENT

RESPONSE

(a) We have checked the IRB protocols of this study and found no restrictions regarding sharing of qualitative in-depth interview data.

RESPONSE

(b) We will upload the final (analyzed) data set in excel to the submission system as a supplementary information file.

EDITOR’S COMMENT

3. We note that this data set consists of interview transcripts. Can you please confirm that all participants gave consent for the interview transcript to be published?

If they DID provide consent for these transcripts to be published, please also confirm that the transcripts do not contain any potentially identifying information (or let us know if the participants consented to having their personal details published and made publicly available). We consider the following details to be identifying information:

- Names, nicknames, and initials

- Age more specific than round numbers

- GPS coordinates, physical addresses, IP addresses, email addresses

- Information in small sample sizes (e.g. 40 students from X class in X year at X university)

- Specific dates (e.g. visit dates, interview dates)

- ID numbers

Or, if the participants DID NOT provide consent for these transcripts to be published:

- Provide a de-identified version of the data or excerpts of interview responses

- Provide information regarding how these transcripts can be accessed by researchers who meet the criteria for access to confidential data, including:

a) the grounds for restriction

b) the name of the ethics committee, Institutional Review Board, or third-party organization that is imposing sharing restrictions on the data

c) a non-author, institutional point of contact that is able to field data access queries, in the interest of maintaining long-term data accessibility.

d) Any relevant data set names, URLs, DOIs, etc. that an independent researcher would need in order to request your minimal data set.

For further information on sharing data that contains sensitive participant information, please see: https://journals.plos.org/plosone/s/data-availability#loc-human-research-participant-data-and-other-sensitive-data

If there are ethical, legal, or third-party restrictions upon your dataset, you must provide all of the following details (https://journals.plos.org/plosone/s/data-availability#loc-acceptable-data-access-restrictions):

1. A complete description of the dataset

2. The nature of the restrictions upon the data (ethical, legal, or owned by a third party) and the reasoning behind them

3. The full name of the body imposing the restrictions upon your dataset (ethics committee, institution, data access committee, etc)

4. If the data are owned by a third party, confirmation of whether the authors received any special privileges in accessing the data that other researchers would not have

5. Direct, non-author contact information (preferably email) for the body imposing the restrictions upon the data, to which data access requests can be sent

RESPONSE

We confirm that all participants gave verbal informed consent prior to data collection. We also confirm that the transcripts do not contain any potentially identifiable information (i.e., names and derivatives of names, specific age, specific date). We also redacted all circumstantial information that may allow for indirect identification of the participants or locations from the study findings. We did not collect ID numbers, GPS coordinates, physical addresses, IP addresses, or email addresses.

Additional Editor Comments:

EDITOR's COMMENT

4. Use the COREQ checklist for qualitative interviews and attach it as a supplementary document.

RESPONSE

We thank the editor for the suggestion, and we have uploaded the COREQ checklist for our study as a supplementary information document accordingly.

EDITOR's COMMENT

5. Introduction: It would be better to explain the COVID-19 waves and the epidemic situation in Thailand for clearer understanding to the readers. Was data collection conducted during the peak of the epidemic, or before or after a major wave?

RESPONSE

We thank the editor for the suggestion. We added about the COVID-19 waves and the epidemic situation in Thailand to the INTRODUCTION section as follows (Line 69-72):

“Thailand experienced five distinct waves of COVID-19 between 2020 and 2022, where the fifth wave of COVID-19 occurred in late 2022, and included the emergence of more transmissible variants like Alpha and Delta [10]. As of 13 April 2024, Thailand had a cumulative total of 4,770,149 confirmed cases with 34,586 deaths [11].”

EDITOR's COMMENT

6. Introduction: Information regarding the importance of adhering to COVID-19 protective behaviors and vaccination is missing. RESPONSE

We thank the editor for the suggestion. We added about information regarding the importance of adhering to COVID-19 protective behaviors and vaccination to the INTRODUCTION section as follows (Line 79�83):

“Public health interventions to prevent and control COVID-19 have included hand hygiene, face covering use, social distancing in public places, isolation, travel restrictions, vaccination, etc. [22–25]. Different interventions affect disease transmission in different ways [26–28], and a comprehensive strategy is deemed needed in order to effectively control COVID-19 infection [29].”

EDITOR's COMMENT

7. Were the migrant workers you collected data from legal or undocumented workers? Legal and undocumented migrant workers may face different barriers.

RESPONSE

We allowed both types of migrants. But the migrants that we interviewed were all documented. We realize that such circumstances also limited the generalizability of our findings and have added the following remarks to the DISCUSSION section (in the “Strengths and Limitations” sub-section) as follows (Line 457-461):

“Firstly, although our protocols did not exclude undocumented migrants, we were only able to recruit documented migrants working in manual labor occupations common in southern Thailand. These circumstances limited the generalizability of our study findings. The barriers among undocumented migrants might have been vastly different from those reported by our participants.”

EDITOR's COMMENT

8. I am curious about the accuracy of Google Translate in translating Burmese and Thai texts into English. Did the study team review and correct the translations after using the tool?

RESPONSE

Yes. HH, a native speaker of Burmese, checked the accuracy of the translation and corrected the mistakes manually. WW, a native speaker of Thai, also performed similar procedures for the transcripts of interviews with the employers and healthcare workers.

We have added the following remarks to the METHODS section (in the "Data management and analysis" sub-section) as follows (Line 202-203):

“HH then manually checked the translations and corrected the parts deemed to be inaccurate.”

We have added the following remarks to the METHODS section (in the "Data management and analysis" sub-section) as follows (Line 207-208):

“A co-investigator and the corresponding author (WW) who spoke Thai as a native language, manually checked the translations and corrected the parts deemed to be inaccurate.”

EDITOR's COMMENT

9. Provide the background characteristics of the participants for all three categories.

RESPONSE

We have added the table of background characteristics of study participants in the RESULTS section in Table 1. (Page 13).

EDITOR's COMMENT

10. Currently, the investigators have analyzed the information based on the type of participant. Please consider analyzing the data by theme, covering all stakeholders. For example, under the theme of health education barrier, presenting perspectives from migrant workers, employers, and healthcare providers would make the findings easier to understand.

RESPONSE

We thank the reviewer for the comment. The questions were drafted separately for each stakeholder group and the probing differed by groups. Therefore, data may not be completely comparable between groups.

EDITOR's COMMENT

11. Move Tables 3 and 4 to the supplementary section.

RESPONSE

We thank the editor for the comment. We have changed Table 3 into Supplementary Table 1 and changed Table 4 into Supplementary Table 2.

########################################

Reviewers' comments:

########################################

Reviewer #1:

REVIEWER’s COMMENT

Thank you very much for your efforts in adding new information on perspectives regarding COVID-19 protective behaviors and vaccination among Myanmar migrant workers, Thai health care providers and Thai employers. In the attachment, I included my recommendations.

RESPONSE

We thank the reviewer for the comments, and we have attempted to address them on a point-by-point basis.

Introduction

REVIEWER’s COMMENT

1. Line 65-67: The statement mentions that 1.65 million migrants from Myanmar are in Thailand. It will be better to clarify the year form which this data was sourced.

RESPONSE

We thank the reviewer for the comment. We have updated the statistics of Myanmar migrant workers in the INTRODUCTION section as follows (Line 65-68):

“The majority of migrants in Thailand come from neighboring Myanmar with 1.65 million registered workers in 2021 according to the statistics from the Department of Employment, Ministry of Labour, Thailand, [8], most of whom work in low-skilled jobs such as construction, factories, and fisheries [9].”

REVIEWER’s COMMENT

2. Please consider adding information about the general health-seeking behaviors of migrants in the study area, especially focusing on irregular migrants. This could provide insights into their specific challenges in accessing COVID-19 preventative measures.

RESPONSE

We thank the reviewer for the comment. We have added the information about general health-seeking behaviors of migrants from previous studies in the study area in the INTRODUCTION section as follows (line 92-95):

“These challenges are further compounded by previous studies conducted in Southern Thailand, which highlighted unsatisfactory health-seeking behaviors among Myanmar seafarers in Pattani province [32], as well as factory, construction, and rubber trapping workers in Songkhla province [33].”

Methods

REVIEWER’s COMMENT

3. Line 98-99: It would be beneficial to include information on the non-pharmacological COVID-19 preventative interventions and restrictions in place during the study period, as these had evolved over time.

RESPONSE

We thank the reviewer for the comment. We have added information on non-pharmacological COVID-19 protective measures and restrictions in our study area in the METHODS section (“Study Design and Setting” sub-section) as follows (Line 115-120):

“During the COVID-19 pandemic (2020–2022), the industries in Southern Thailand were transiently affected, and the government issued several non-pharmacological measures, including social distancing, compulsory mask wearing, travel restrictions, and workplace disinfection, to prevent further spread among the workers. In late 2021, the government offered vaccinations to migrant workers free of charge as part of an inclusive vaccine policy [35]. ”

REVIEWER’s COMMENT

4. Line 99-101: It might be useful to explain why these cities were chosen for the study. It would be better to describe the number of migrant workers in those cities. I checked the cited reference, and it is in Thai language only.

RESPONSE

We thank the reviewer for the comment. We have added information about the rationale on choosing the study setting and the number of Myanmar migrant workers in that area according to the statistics from the Department of Employment, Ministry of Labour, Thailand, in the METHODS section (“Study Design and Setting” sub-section) as follows (Line 121-128):

“Study areas included the cities of Hat Yai city of Songkhla province and Pattani city of Pattani province in southern Thailand. Songkhla and Pattani provinces are major industrial commercial area of the Southern Thailand with many factories that process rubber, wood and seafood as well as numerous construction and fishery sites [36]. Many Myanmar migrant laborers live in these provinces, with different ethnic groups, religions, and cultures. According to the 2021 Thailand Employment Statistics, there were 19,810 and 4,122 legal Myanmar migrant workers granted by cabinet resolution in Songkhla and Pattani provinces, respectively [37].”

REVIEWER’s COMMENT

5. Line 116-118: It was mentioned that purposive sampling was used. Providing more detail on how key informants and participants were identified could be more informative.

RESPONSE

We thank the reviewer for the comment. We have added detail information on detail on how key informants and participants were identified in the METHODS section (“Study participants” sub-section) as follows (Line 145-153):

“Myanmar migrants were identified with the help of the key informants from the Migrant Workers Right Network (MWRN) and the Stella Maris Organization. These key informants were Myanmar Nationalities who had been working in the study area for more than 10 years. They could well acquaint with the Myanmar migrant population within Hat Yai and Pattani and could provide information on places of worksite and residence, types of occupations, legal status, approximate population sizes, and their lifestyles. For Thai employers and Thai healthcare providers, they were also identified by the Thai research coordinator and invited via official request letters, emails, or phone calls and invited to participate in individual in-depth interviews.”

REVIEWER’s COMMENT

6. It will be better to explain how cultural nuances were handled during translation and interviews.

RESPONSE

We thank the reviewer for comment. We assigned Thai reviewers to

---

## [Decision Letter · Decision Letter 1]

3 Jan 2025

Perspectives of stakeholders on barriers to COVID-19 protective behaviors adherence and vaccination among Myanmar migrant workers in southern Thailand: a qualitative study

PONE-D-24-34969R1

Dear Dr. Wichaidit,

We’re pleased to inform you that your manuscript has been judged scientifically suitable for publication and will be formally accepted for publication once it meets all outstanding technical requirements.

Kind regards,

Kyaw Lwin Show, MPH, PhD

Academic Editor

PLOS ONE

Additional Editor Comments (optional):

Reviewers' comments:

Reviewer's Responses to Questions

**Comments to the Author**

Reviewer #1: All comments have been addressed

Reviewer #2: All comments have been addressed

2. Is the manuscript technically sound, and do the data support the conclusions?

Reviewer #1: Yes

Reviewer #2: Partly

3. Has the statistical analysis been performed appropriately and rigorously?

Reviewer #1: Yes

Reviewer #2: N/A

4. Have the authors made all data underlying the findings in their manuscript fully available?

Reviewer #1: Yes

Reviewer #2: Yes

5. Is the manuscript presented in an intelligible fashion and written in standard English?

Reviewer #1: Yes

Reviewer #2: Yes

Reviewer #1: Thank you for addressing all the comments and recommendations provided in the first review. The reviewer acknowledges the significant improvements made throughout the manuscript compared to the original version. Additionally, the inclusion of the requested information has greatly enhanced the context and clarity of the study, facilitating a better understanding of its scope and findings.

Reviewer #2: Thank you for addressing all the comments. Please ensure consistency in word usage throughout the manuscript. I have observed that both "undocumented migrant" and "irregular migrant" are being used. While I understand that both terms convey the same meaning, kindly use only one consistently throughout the manuscript.

Congratulations to all the authors for this important article.

**Do you want your identity to be public for this peer review?** For information about this choice, including consent withdrawal, please see our Privacy Policy

Reviewer #1: No

Reviewer #2: No

---

## [Editor Report · Acceptance letter]

PONE-D-24-34969R1

PLOS ONE

Dear Dr. Wichaidit,

I'm pleased to inform you that your manuscript has been deemed suitable for publication in PLOS ONE. Congratulations! Your manuscript is now being handed over to our production team.

Kind regards,

on behalf of

Dr. Kyaw Lwin Show

Academic Editor

PLOS ONE